# Generation and Validation of an Anti-Human PANK3 Mouse Monoclonal Antibody

**DOI:** 10.3390/biom12091323

**Published:** 2022-09-19

**Authors:** Sunada Khadka, Long Vien, Paul Leonard, Laura Bover, Florian Muller

**Affiliations:** 1Department of Cancer Systems Imaging, The University of Texas MD Anderson Cancer Center, Houston, TX 77054, USA; 2Department of Cancer Biology, The University of Texas MD Anderson Cancer Center, Houston, TX 77030, USA; 3MD Anderson UT Health Graduate School of Biomedical Sciences, Houston, TX 77054, USA; 4Department of Immunology, The University of Texas MD Anderson Cancer Center, Houston, TX 77054, USA; 5Institute of Applied Cancer Science, The University of Texas MD Anderson Cancer Center, Houston, TX 77054, USA; 6Department of Genomics Medicine, The University of Texas MD Anderson Cancer Center, Houston, TX 77054, USA

**Keywords:** pantothenate kinases, PANK3, monoclonal antibody, CoA

## Abstract

Coenzyme A (CoA) is an essential co-factor at the intersection of diverse metabolic pathways. Cellular CoA biosynthesis is regulated at the first committed step—phosphorylation of pantothenic acid—catalyzed by pantothenate kinases (PANK1,2,3 in humans, PANK3 being the most highly expressed). Despite the critical importance of CoA in metabolism, the differential roles of PANK isoforms remain poorly understood. Our investigations of PANK proteins as potential precision oncology collateral lethality targets (*PANK1* is co-deleted as part of the *PTEN* locus in some highly aggressive cancers) were severely hindered by a dearth of commercial antibodies that can reliably detect endogenous PANK3 protein. While we successfully validated commercial antibodies for PANK1 and PANK2 using CRISPR knockout cell lines, we found no commercial antibody that could detect endogenous PANK3. We therefore set out to generate a mouse monoclonal antibody against human PANK3 protein. We demonstrate that a clone (Clone MDA-299-62A) can reliably detect endogenous PANK3 protein in cancer cell lines, with band-specificity confirmed by CRISPR PANK3 knockout and knockdown cell lines. Sub-cellular fractionation shows that PANK3 is overwhelmingly cytosolic and expressed broadly across cancer cell lines. PANK3 monoclonal antibody MDA-299-62A should prove a valuable tool for researchers investigating this understudied family of metabolic enzymes in health and disease.

## 1. Introduction

Coenzyme A (CoA) is a central co-factor for more than 4% of known cellular enzymes and is involved in hundreds of biochemical reactions in mammalian cells [1]. As an important acyl group carrier, CoA is indispensable for reactions of the central metabolic pathways, such as the synthesis and oxidation of fatty acids; oxidation of pyruvate in the TCA cycle; synthesis of lipids, isoprenoids, and sterols; amino acid metabolism; and porphyrin synthesis [1,2]. Additionally, the CoA-derived phosphopantetheine prosthetic group is also critical for enzymes involved in fatty acid, nonribosomal peptide, and polyketide synthesis [1,3,4]. Pantothenate kinases, a family of four evolutionarily conserved proteins, regulate the de-novo CoA biosynthesis in prokaryotes as well as eukaryotes [5]. The family of PANKs constitutes four catalytically active kinases: nuclear PANK1α and cytosolic PANK1β (both encoded by *PANK1*); mitochondrial (intermembrane space) PANK2; cytosolic PANK3; and a phosphatase—cytosolic PANK4 [6,7]. As the rate-limiting enzymes of the pathway, PANK1, 2, and 3 phosphorylate pantothenic acid (vitamin B_5_), which is converted into CoA in a series of four evolutionarily conserved enzymatic reactions, while PANK4 negatively regulates CoA synthesis by dephosphorylation of the pathway metabolites [6,7,8]. PANK isoforms are also differentially expressed and regulated, enabling these proteins to sense and maintain the levels of CoA and its thioesters differentially in a specific cellular compartment [2,9]. PANK proteins have been studied extensively in the context of normal mammalian physiology and pathologies that arise due to their dysregulation [10,11,12,13,14]. Inactivating mutations in PANK2 protein have been linked to an inborn error of CoA metabolism, clinically manifesting as a neurodegenerative condition called pantothenate kinase-associated neurodegeneration (PKAN) in humans [15,16,17]. This pivotal finding has accelerated novel research efforts to restore PANK function to correct the CoA deficiency-induced neurodegeneration [15,18,19,20,21]. Similarly, cancers with homozygous (bi-allelic) deletion of the tumor suppressor *PTEN*, which are typically highly aggressive, have a poor prognosis, and are refractory to treatment [22,23,24], can show co-deletion of the neighboring gene *PANK1*, making the redundant PANK proteins potentially attractive therapeutic targets in these cancers [25,26]. Despite the obvious importance of PANKs in CoA biosynthesis, the lack of reliable antibodies that can detect and distinguish endogenous PANK isoform proteins, has significantly thwarted efforts to better understand their role in both normal physiology and in pathological contexts, as well as to characterize them as therapeutic targets. Generation of antibodies against PANK proteins, especially PANK3, has been significantly challenging given the extreme homology between the typical host species, such as mice/rabbits, and humans. Additionally, the structural similarities in the PANK isozymes pose additional challenges in generating isozyme-specific antibodies. While many commercial antibodies may detect unphysiological levels of overexpressed PANK proteins, few antibodies are able to detect endogenously expressed PANKs. Through an extensive and laborious screening of commercially available antibodies, we validated the antibodies for PANK1 and PANK2 for the detection of endogenous proteins using PANK-specific CRISPR KO cell lines. However, we could not find commercial antibodies capable of detecting endogenous human PANK3 protein.

Here, we describe the generation, optimization, and validation of a mouse monoclonal antibody (mAb) (MDA-299-62A) against human PANK3 protein. We demonstrate that MDA-299-62A can reliably detect purified recombinant human PANK3 and endogenous PANK3 protein in cancer cell lysates, with the band specificity validated by PANK3 CRISPR knockout cancer cell lines. The PANK3 antibody generated in this study could serve as a reliable tool for quantifying endogenous PANK3 protein and should prove to be valuable for studies investigating these enzymes in health and disease. 

## 2. Methods

### 2.1. Protein Expression and Purification

The pET28a plasmid vector containing DNA sequences encoding the PANK3 protein (residues pro12 to Asn368) was purchased from Addgene, USA (25518) and transformed into both *E. coli* BL21 (DE3) and *E. coli* Rosetta2 (DE3) strains. The *E. coli* cells were grown in Terrific Broth media at 37 °C until the optical density at 600 nm reached 0.5 absorbance units. The temperature of the culture was then reduced to 18 °C prior to induction of the recombinant protein expression by the addition of 1 mM isopropyl β-D-1-thiogalactopyranoside (IPTG). All protein purification steps were performed at 4 °C. After harvesting the cells by centrifugation, the *E. coli* cells were lysed by re-suspending the pellet in 10 mM Tris pH 7.5, 0.5 M NaCl, 5% glycerol, 5 mM Imidazole, protease (cOmplete^TM^ mini, Roche, USA #11836153001) and phosphatase inhibitors (PhosSTOP^TM^, Roche, USA #5892970001), 30 µg mL^−1^ DNase, and 500 µg mL^−1^ lysozyme at pH 7.5, and lysed by sonication. The lysate was centrifuged at 20,000 rpm for 45 min to remove insoluble material. The recombinant protein was purified from the clarified lysate using a 2 mL Ni-NTA column pre-equilibrated in 10 mM Tris pH 7.5, 0.5 M NaCl, 5% glycerol, and 5 mM imidazole buffer. The Ni-NTA column was washed with 10 mM Tris-HCl, 0.5 M NaCl, 5% glycerol, and 30 mM imidazole at pH 7.5. The bound proteins were eluted at 10 mM Tris-HCl, 0.5 M NaCl, 5% glycerol, and 250 mM Imidazole at pH 7.5. The protein fractions eluted were assessed by SDS PAGE.

### 2.2. Monoclonal Antibody Production and Purification

All animal experiments were performed using protocols approved by the Institutional Animal Care and Use Committee (IACUC). Antibody production using hybridoma technology and purifications were done at the MD Anderson Monoclonal Antibody Core. A detailed report on the protocol has been described [27]. Briefly, purified recombinant PANK3 protein was emulsified using the incomplete Freund Adjuvant (IFA, InvivoGen, San Diego, CA, USA) in a 1:1 ratio. Two NZBWF1/J mice and one BALB/c mouse were used for immunization. In total, 10 µg of recombinant protein (20 µL volume total) was administered through the footpad route every two days for the first five injections, and the dose was raised to 15 µg for the three booster injections, administered weekly. After completion of immunization, on Day 31, mice were sacrificed and the popliteal lymph nodes and spleen from the immunized mice with the highest serum titer were harvested using sterile methods. B cell isolated either from the lymph node or the spleen were fused with Sp20 murine myeloma cells and selected in hypoxanthine–aminopterin–thymidine (HAT) medium, and single cell clones were identified and propagated. Enzyme-linked immunoassay (ELISA) screening was performed using the supernatant from each positive hybridoma clone to determine the efficiency of the target protein binding by the mAb-containing supernatants. Western blot was performed on HeLa cells to identify the clones that could detect endogenous PANK3 protein. The selected hybridoma clones were expanded and 300 mL of mAb-rich supernatants were collected and filtered using a 0.45 µm membrane. The supernatants were added to Protein A columns for affinity chromatography purification. The antibodies captured in the column were eluted using an elution buffer (0.1 M glycine-HCl, pH = 3) that was neutralized with a 1 M Tris-HCl buffer. Overnight dialysis was performed in PBS to concentrate and preserve the antibody activity in a neutral buffer.

### 2.3. ELISA Assay

ELISA plates were first coated with 100 µL of coating buffer containing the recombinant target protein used for immunization at 1 µg/mL and incubated at 4 °C overnight. After 3X PBST wash, the plate was blocked with 100 µL of ELISA blocking buffer at RT for 1 h. In total, 50 µL of diluted serum from the immunized mice or the media supernatant from each hybridoma clone, which serve as the primary antibody, were added to individual wells, respectively, in the corresponding screenings and incubated for 1 h. The plate was then washed 3X in PBST, and 50 µL of goat anti-mouse HRP conjugated secondary antibody was added to each well and incubated for one hour. After 4X PBST washes, horseradish peroxidase (HRP) substrate TMB was added and the reaction was stopped by addition of H_2_SO_4_ after 30 min. The OD 450 nm values were then measured using a spectrophotometer. 

### 2.4. Cell Culture

The cell lines used in this work were HeLa (CVCL_0030, Cervical Carcinoma), 537 MEL (CVCL_8052, Melanoma), HAP1(CVCL_YO91), SKMEL-5 (CVCL_0527, Melanoma), SKMEL28 (CVCL_0526, Melanoma), SKMEL2 (CVCL_0069, Melanoma), G59 (CVCL_N729, Glioblastoma), D423-MG (CVCL_1160, Glioblastoma), referred to as D423 in the figures, LN319, a sub-clone of LN-99267, (CVCL_3958, Glioblastoma), D502 (CVCL_1162, Glioblastoma), and HEK293T (CVCL_0063). The cell lines were authenticated at the MD Anderson Cytogenetics and Cell Authentication Core. The cell lines were obtained from the following sources: HeLa (ATCC), 537 MEL (NCI), HAP1 (Horizon Discovery), SKMEL-5 (ATCC), SKMEL-2 (ATCC), SKMEL-28 (ATCC), G59 (Prof.Dr. Katrin Lamszus, Universitätsklinikum Hamburg-Eppendorf), D423 (Darrel Bigner, Duke University), LN319 (ATCC), D502 (ATCC), and HEK 293T(ATCC). All cells were maintained in RPMI medium with 2 mM glutamine (Cellgro/Corning #10-040-CV) supplemented with 10% FBS (Gibco/Life Technologies, USA #16140-071) and 1% pen-strep (Gibco/Life Technologies#15140-122). 

### 2.5. Generation of CRISPR Knockout Clones

CRISPR knockout was performed using the Santa Cruz dual plasmid CRISPR system. PANK CRISPR plasmids (PANK1 (sc-408890), PANK2 (sc-405120), and PANK3 (sc-409325)) and PANK HDR plasmids (PANK1 (sc-408890-HDR), PANK2 (sc-405120-HDR), and PANK3 (sc-409325-HDR)) were purchased from Santa Cruz Biotechnologies. In total, 1 µg of PANK plasmids and 1 µg of PANK HDR plasmids were mixed in 140 µL plasmid transfection medium (sc-108062). In a separate tube, 5 µL of UltraCruz^®^ (sc-395739) transfection reagent was added to 145 µL transfection medium. Plasmid and transfection reagents were mixed and incubated for 10 min at RT. The cell culture medium in the target cells was replaced with fresh medium and the CRISPR/HDR plasmid mix was slowly added to the plates, and the plates were gently shaken to allow for homogenous mixing. Then, 24 h later, the medium was replaced, and the cells subjected to 2 µg/mL puromycin selection for a week. The cells were then trypsinized, washed in PBS, and resuspended in FACS buffer (1%FBS + 0.5 mM EDTA in PBS). Single cell sorting was performed with an RFP channel as a gate using a BD FACS ARIA Cell Sorter. Confirmation of knockout was done by Western blot. 

### 2.6. Generation of Stable Dox Inducible PANK3 shRNA Cell Lines

Third-generation lentivirus packaging system was used to generate stable doxycycline-inducible shRNA-expressing cells. Recombinant lentiviral particles were generated by transient transfection of HEK 293T cells following a standard protocol. Plasmids (pRSIT17H-U6Tet-sh-CMV-TetRep-2A-TagGFP2-2A-Hygro) encoding a doxycycline-inducible shRNA, hygromycin-resistance marker, and red fluorescent protein (RFP) were purchased from Cellecta. For transfection, 10 µg of the shRNA plasmid was mixed with 5 µg of pMd2g, 5 µg of pREV, 5 µg of p8.74, and 10 µg of shRNA vector, and the plasmids were transfected into HEK 293T cells plated in 100 mm^2^ dishes. The medium was replaced with a fresh one after 12 h. Viral supernatants were collected, and fresh medium was added to the HEK 293T cells every 24 h for the first 48 h after the initial medium change. Cell debris were eliminated by centrifugation at 500× *g*, and the supernatant containing the virus particles was sterile filtered through a 0.45 µm filter. To transduce the target cells, the filtered viral particles were mixed with an equal volume of fresh medium and polybrene (8 µg/mL) and added to the target cells. The medium on the target cells was replaced 24 h after infection. Then, 48 h after infection, 200 µg /mL hygromycin was added for selection. Knockdown of PANK3 was induced by addition of 2 µg/mL dox for 48–72 h and knockdown were confirmed by Western blot. 

### 2.7. Western Blot

#### 2.7.1. Whole Cell Lysate Preparation

Cells were grown in 6-well plates for 48–72 h and lysates were harvested by washing the cells twice with ice-cold phosphate-buffered saline (PBS). An ice-cold RIPA buffer with protease (cOmplete™ mini, Roche, USA #11836153001) and phosphatase inhibitors (PhosSTOP^TM^, Roche, USA #5892970001) was added and the samples were then sonicated. 

#### 2.7.2. Cytosolic and Mitochondrial Sub-Cellular Fractionation 

Sub-cellular fractionation was performed using a Thermo Fisher Scientific (USA) sub-cellular fractionation kit (#78840), according to the manufacturer’s directions. Briefly, cells were trypsinized and pellets were washed with ice-cold PBS. Ice-cold cytoplasmic extraction buffer with 1X protease inhibitor cocktail was added to re-suspend the pellet and the sample was incubated on ice for 10 min with gentle mixing. The cell lysate was then centrifuged at 500× *g* for 5 min at 4 °C, and the cytosolic fraction in the supernatant was transferred to a pre-chilled tube, without agitating the pellet. Ice-cold membrane extraction buffer was added to the pellet, and the tube was vortexed at the highest setting for 5 s and incubated on ice for 10 min with gentle mixing. The tube was then centrifuged at 3000× *g* for 5 min, and membrane fraction was isolated from the supernatant. 

A BCA assay (Thermo Fisher Scientific, USA #23227) was used to determine the protein concentration. Proteins were separated by Nu-PAGE SDS-PAGE (4–12% gradient) and transferred onto nitrocellulose membranes using the semi-dry method (Trans-Blot® Turbo^TM^ Transfer System, USA #1704150). Effective transfer of proteins onto the membrane was verified with Ponceau S staining. Then, 5% non-fat dry milk in tris-buffered saline (TBS) with 0.1% Tween 20 (TBST) was used as the blocking agent to block the non-specific sites on the membrane. Primary antibodies (PANK1 Cell Signaling Technologies, USA, 1:1000; PANK2 Origene, USA, 1:1000; PANK3 5 µg/mL, TPI1 ProteinTech, USA #10713-1-AP 1:10,000) were added to the membranes and incubated overnight at 4 °C with gentle rocking. On the second day, membranes were washed 3X for 5 min with TBST. Membranes were then incubated in HRP tagged secondary antibody (1:5000) for 1 h with gentle rocking and then washed 3X for 5 min with TBST. Membranes were incubated first with ECL substrate (Thermo Fisher Scientific #32106) and exposed on X-ray films in a dark room. Exposure times were varied as necessary. For especially the faint bands, (Thermo Fisher Scientific (USA) SuperSignal West Femto #4096) was used as a substrate before X-ray exposure. 

## 3. Results

### 3.1. CRISPR-Validated Commercial Antibodies for PANK1 and PANK2, but Not PANK3

To identify commercially available antibodies that can detect endogenous PANK proteins, we tested antibodies from diverse commercial suppliers. We found many commercial antibodies can detect highly overexpressed unphysiological levels of PANKs, such as those obtained by transient transfection of PANK overexpressing plasmids. However, very few can specifically detect endogenous expression of PANKs. Additionally, the band specificities of these antibodies were not validated with a proper negative control by using PANK-deficient or PANK CRISPR knockout cell lines. To mitigate these issues, we purchased *PANK1*, *PANK2*, or *PANK3* CRISPR knockout (KO) clones in HAP1 cells (Figure 1A, Appendix A) from Horizon Discovery. HAP1 is a near-haploid cell line derived from KBM-7, a chronic myelogenous leukemia cell line. Due to its haploidy, HAP1 is an ideal model for genetic manipulations with CRISPR, and holds great potential for genetic screening studies [28]. Since HAP1 cells have almost one copy of most genes, CRISPR-induced heterozygous mutation artifacts (as in a diploid cell) are eliminated, allowing for easy and efficient generation of knockout clones. PANK isoform knockouts in HAP1 cells were initially custom-made to order, but they are now available off-the shelf for other investigators to use. Additionally, we also independently generated *PANK1*, *PANK2*, or *PANK3* knockout clones in multiple different cancer cell lines to validate the band specificity and serve as a negative control for the antibodies (Figure 1C–E, Appendix A). We found that PANK1 CST#23887S rabbit monoclonal antibody detected a band at 50 kD, which corresponds to PANK1β (expected 41.6 kD). We found that the band corresponding to PANK1β disappeared in two independent CRISPR knockout clones in HAP1 and HeLa cells (Figure 1A,C). Further validation of CST PANK1 antibody was also done in *PANK1* endogenous genomic deleted cell lines, such as 537 MEL and G59, and other *PANK1* intact cell lines, which further reinforced the band specificity of the antibody (Figure 1B). We also found that CST #23887S can detect both PANK1α and PANK1β in cells ectopically overexpressing the full length PANK1 cDNA (data not shown). In the cell lines that we employed in this study, we found that endogenous PANK1α expression was minimal compared to PANK1β. 

Similarly, we found that Origene’s #TA501321 mouse monoclonal antibody against PANK2 yields a band at the expected molecular weight (48 kDa for mature PANK2), which disappeared in the *PANK2* CRISPR knockout clones in HAP1, HeLa, and LN319 cells (Figure 1A,E,F). However, we were unable to validate the loss of the bands of PANK1 and PANK2 proteins in the CRISPR KO clones by the ProteinTech antibody (data not shown). We also tested multiple commercially available antibodies against PANK3 using *PANK3* knockout CRISPR cell lines as negative controls. However, none of these antibodies gave the correct size band, and did not show loss of the band in the CRISPR *PANK3* KO cell lines. This led us to invest in generating our PANK3 antibody. 

### 3.2. Production and Purification of His-Tagged Recombinant PANK3 Protein

PANK3 is a 41 kD cytosolic protein that is ubiquitously expressed in all cell types. PANK3 protein is highly homologous between humans and rodents (homology with mice—99.1%; homology with rabbit—100%) (Appendix A), which necessitated the use of the full length human PANK3 protein as the antigen, to allow for multiple and maximal antigenic as well as conformational epitope recognition in mice. The pET28a plasmid encoding six histidine residues fused to 364 amino acid residues of the human PANK3 protein (residues pro12 to Asn368) was used as a vector for recombinant PANK3 production (Figure 2A–C). pET28a plasmid was transformed into *E. coli* and His-PANK3 synthesis was induced by the addition of IPTG into the growth medium. The clarified lysates from the *E. coli* cells were purified using a Ni-NTA column, and the eluted fractions were assessed for the presence of the recombinant PANK3 protein by Coomassie staining of the SDS-PAGE (Figure 2A,C,D). An unidentifiable band was observed around 75 kDa, so the Ni-NTA column elutes were further subjected to size-exclusion gel chromatography to obtain a purified recombinant PANK3 homodimer (Figure 2E, Appendix A). 

### 3.3. Generation and Validation of an Anti-hPANK3 Mouse mAb

To generate a human PANK3-specific antibody, we immunized three mice—two NZB mice and a BALB/c mouse—with recombinant PANK3 protein (Figure 3A). Due to a 99.1% sequence homology in PANK3 protein between a human and mouse (Appendix A), we included the autoimmune NZB mouse model along with the wildtype BALB/c mouse for immunization. The mice were administered five doses of PANK3 antigen mixed 1:1 with IFA every 2 days for the first 15 days, and then the final three booster doses were administered every week (Figure 3A). The serum samples collected from the mice were subjected to ELISA assay for quantification of serum PANK3 antibody titer. Due to the high PANK3 antibody titer in NZB mouse #2, we generated hybridomas using the plasma cells isolated from this mouse (Figure 3B,C). We fused the splenocytes isolated from the immunized mouse with the murine myeloma cells Sp20 and selected the positive hybridoma clones in the HAT medium (Figure 3C). 

To determine the activity and efficacy of each hybridoma clone, we performed ELISA assays using the media supernatant from the hybridoma clones (Figure 4A,B). We further tested the efficacy and specificity of each clone-derived supernatant to detect endogenous PANK3 protein in HeLa cell lysates (Figure 4C, Appendix A). Consistent with the ELISA results, the supernatants from most clones were able to recognize the recombinant antibody by immunoblotting assay (Figure 4C). In whole cell lysates, some of the clones were able to detect a band approximately at the expected molecular weight (~41 kDa); however, none of the clones yielded a clean and specific band for the endogenous PANK3 protein. We chose four clones that optimally recognized the endogenous PANK3 protein for further purification. As evidenced in Figure 4C, all four clones were able to detect the recombinant PANK3 protein, but the sub-optimal specificity for endogenous PANK3 and the presence of many non-specific bands posed significant challenges in further application of the antibody. To mitigate these issues, we ventured on optimizing the antibody and the immunoblotting assay conditions. 

### 3.4. Method-of-Use Optimization of MDA-299-62A Anti-Human PANK3 Mouse mAb

PANK3 is a cytosolic protein in eukaryotes (Figure 5A, Appendix A). To improve the specificity of the antibody in immunoblotting assay, we performed subcellular fractionation to separate the cytosolic proteins, including PANK3, from the total cell lysate. We found that separating the cytosolic proteins by sub-cellular fractionation significantly reduced the non-specific signals and allowed reliable detection of endogenous PANK3 proteins in a broad range of cancer cell lines (Figure 5A,B and Appendix A). Out of the four hybridoma clone supernatants that we purified, clone MDA-299-62A appeared to be the most effective in detecting endogenous PANK3 protein, with the most minimal signal-to-noise ratio. Therefore, we used this clone for further optimization. To conclusively validate that the antibody was indeed recognizing the endogenous PANK3 protein, we employed the CRISPR–Cas9 technology to knock out PANK3 protein in two different cancer cell lines—HAP1 and HeLa (Figure 5B,C, Appendix A). We successfully obtained multiple single-cell PANK3 knockout clones from these cell lines and validated that the antibody detected the correct band (Figure 5B,C, Appendix A). The CRISPR experiment also evidenced that PANK3 is a dispensable gene in most cancer cells, which is consistent with large-scale CRISPR and RNAi screen (DepMap) as well as mouse germline KO studies [2,14,29]. We also performed knockdown studies using a doxycycline-inducible system and found that MDA-299-62A can reliably detect shRNA-mediated knockdown of PANK3 in cancer cells. 

While subcellular fractionation could eliminate the non-specific signals on the immunoblot, it can be a cumbersome experiment to perform for routine use of PANK3 antibody for immunoblotting. Intending to reduce the non-specific signals without the need for subcellular fractionation, we modified the immunoblot assay protocol and employed a longer blocking duration of 6–8 h, compared to the conventional 1 h at room temperature on whole cell lysates (Figure 5E,F, Appendix A). We found that the duration of blocking could significantly dampen the non-specific signals and improved the signal-to-noise ratio in both cytosolic protein fractions as well as whole cell lysates (Figure 5E,F, Appendix A).

## 4. Discussion

CoA is an indispensable cofactor for a myriad of biochemical reactions in cells. PANK proteins control the first and the rate-limiting step in the de-novo coenzyme A biosynthesis. The crucial discovery of the link between PANK2-inactivating mutations and PKAN disease in humans has accelerated investigative efforts to mitigate PKAN-associated pathologies [15,21]. Thus, multiple therapeutic approaches are currently being explored in preclinical and clinical studies to restore CoA levels to combat CoA deficiency in these pathologies [19,20]. Besides PKAN, owing to the essentiality of CoA at the cellular as well as organismal level, PANK and other CoA pathways proteins have also garnered interest as attractive therapeutic targets against parasites such as *Plasmodium falciparum* and *Toxoplasma gondii* [2,29,30,31]. The importance of CoA in microbes and the distinctions in the CoA biosynthesis pathway proteins between microbes and mammals have also propelled investigative interests on CoA biosynthesis as anti-bacterial targets [32,33]. Our drug-target discovery investigations of PANK proteins also relies on the essentiality of PANK proteins for CoA biosynthesis and aims to exploit the redundancies in the PANK proteins to specifically target *PANK1*-deleted cancers, which can be co-deleted as part of the *PTEN* tumor suppressor locus, whose homozygous deletion is associated with poor prognosis, high malignancy, and resistance to both conventional chemotherapy and precision oncology drugs [22,23,24,25,26]. Despite significant investigative interest in understanding the roles of PANK proteins in normal and pathological conditions, the paucity of antibodies to detect endogenous PANK proteins has been a detriment to the advancement of these studies.

In this study, we report the validation of commercially available PANK1 and PANK2 antibodies and demonstrate detection of band specificity using PANK1 and PANK2 CRISPR KO cells. Since we were unable to obtain any reliable PANK3 antibody against endogenous human PANK3 protein, we generated and validated a human PANK3 specific mouse monoclonal antibody. We show that MDA-299-62A can reliably detect recombinant PANK3 protein and endogenous PANK3 protein in ELISA and immunoblotting assays. We confirmed the band specificity of our antibody by using a PANK3 CRISPR knockout cell line. Our results show that the antibody MDA-299-62A can be routinely used for immunoblotting assays, but with the caveat of multiple other unspecific bands along with the PANK3 protein in the cell lysate (though this is also the case with commercial anti-PANK2 mouse mAb (Origene #TA501321, Figure 1E,F)). The use of full-length protein as an antigen possibly explains the detection of many unspecific bands by the antibody. PANK3 is extremely conserved between humans and rodents (rabbit, 100%). So, the antibody was raised in mice, which shares a 99.1% homology to human PANK3 protein. We used the full-length protein in an autoimmune mouse model, to allow for maximum antigen recognition and antibody production. While developing the antibody in organisms with significant sequence differences could alleviate this issue; however, our attempt at developing a PANK3 polyclonal antibody in chickens did not yield improved results (data not shown). 

To alleviate the issue of unspecific bands recognition by MDA-299-62A, we demonstrate that by performing subcellular fractionation, we can concentrate PANK3 in the cytosol fraction, which can significantly minimize unspecific bands. Since performing sub-cellular fractionation on a routine basis can be cumbersome, we demonstrate that by extending the duration of blocking of the membranes, we can significantly minimize these unspecific bands. These mitigation measures are only relevant for immunoblotting assays, and further optimization of MDA-299-62A mAb will be required for immunohistochemistry, immunofluorescence, and fluorescence-activated cell sorting (FACS) experiments. However, the consistent and reliable results that we obtained with MDA-299-62A from well-controlled experiments show that the antibody is robust enough to study the genetic interactions of PANK isoform knockout for the purposes of validating PANK proteins as collateral lethality targets in cancer. 

## Figures and Tables

**Figure 1 biomolecules-12-01323-f001:**
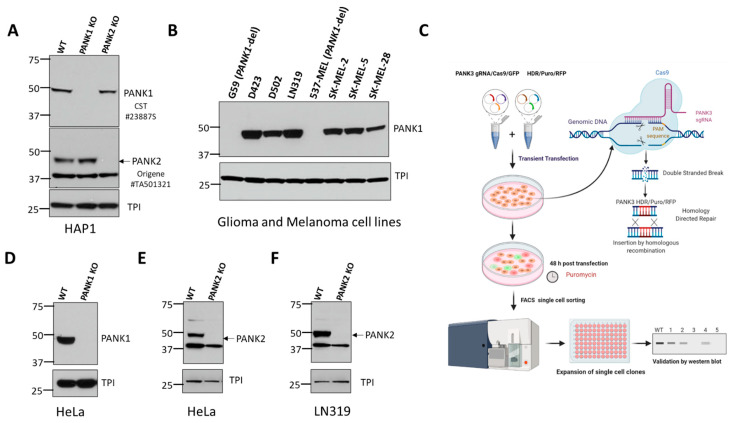
Validation of commercially available antibodies for PANK1 and PANK2. (**A**) Commercially obtained and genomically validated CRISPR knockout HAP1 cell lines were used as negative controls to verify the specificity of the PANK antibodies. Western blot confirming the loss of signal of PANK1 and loss of PANK2 in HAP1 PANK1 and PANK2 CRISPR knockout clones purchased from Horizon Discovery. Rabbit anti-PANK1 mAb from CST (#23887S) correctly identifies a single band at 50 kDa (expected molecular weight of PANK1 protein) in WT HAP1 cells, and the band is absent in the PANK1 KO CRISPR clone. Origene mouse anti-PANK2 (mAb#TA501321) identifies endogenous PANK2 protein in wild type HAP1 cells at the expected size of 48 kD, and the band disappears in the PANK2 CRISPR KO clone. A non-specific band at 37 kD is evident but is sufficiently distinct not to confound interpretation. Triose phosphate isomerase (TPI) was used as a loading control. (**B**) Validation of CST PANK1 antibody using the PANK1 deleted and PANK1 intact glioma and melanoma lines. G59 (glioma) and 537 MEL (melanoma) are PANK1 homozygous deleted cancer cell lines. (**C**) Schematic showing the Santa Cruz two-plasmid (gRNA/Cas9/GFP and HDR/Puromycin/RFP) system-mediated CRISPR KO. (**D**–**F**). Independent in-house generation of PANK1 and PANK2 CRISPR KO in cancer cells, and validation by Western blot.

**Figure 2 biomolecules-12-01323-f002:**
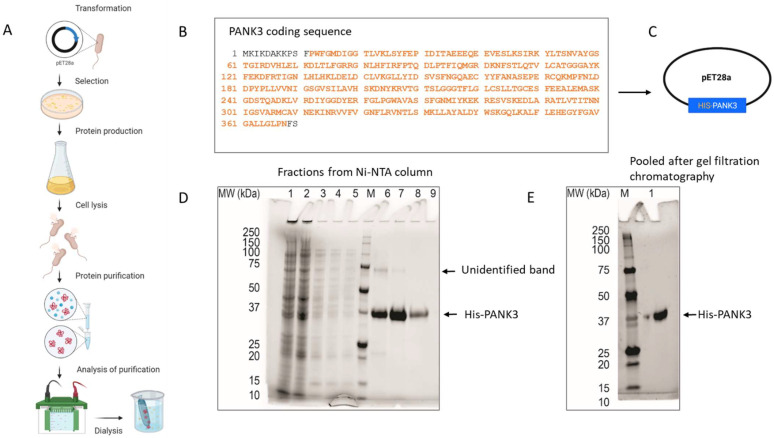
Generation and purification of recombinant PANK3 protein. (**A**–**C**) Schematic showing the recombinant PANK3 production and purification. *E. coli* BL21 (DE3) and *E. coli* Rosetta2 (DE3) strains were transformed with pET28a plasmid vector encoding N-terminal His-tagged fusion PANK3 protein (residues pro12 to Asn368—highlighted in orange). The *E. coli* cells were grown in Terrific Broth medium at 37 °C overnight and then the temperature of the culture was reduced to 18 °C prior to induction of the recombinant protein expression by the addition of 1 mM IPTG. Cells were harvested and lysed, and the soluble supernatant was applied in a pre-equilibrated Ni-NTA column, to allow the histidine tag to bind to the column. After washing the column, the bound proteins were eluted in 10 different fractions. (**D**,**E**) SDS PAGE gel was stained with Coomassie Blue to assess the presence of His-PANK3 in the fractions. PANK3 protein purified from the Ni-NTA column was further subjected to gel filtration chromatography. Purified PANK3 protein eluted through gel filtration was validated by SDS-PAGE gel and Coomassie staining.

**Figure 3 biomolecules-12-01323-f003:**
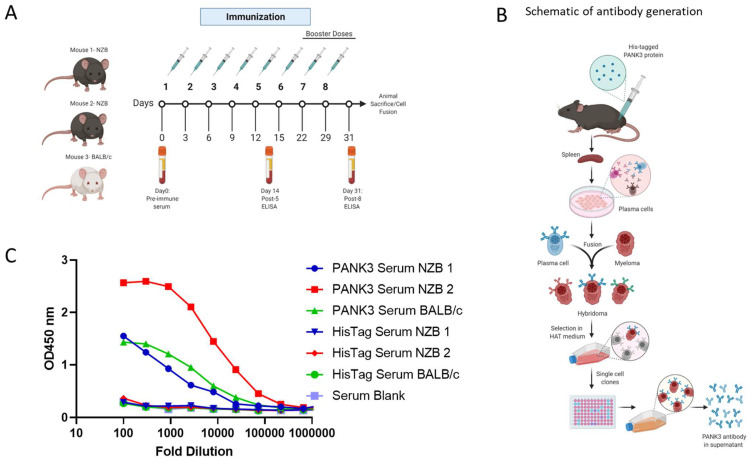
Generation of PANK3 antibody and detection of serum antibody levels by ELISA. (**A**) Schematic showing the schedule of PANK3 protein immunization and blood collection for ELISA. PANK3 was administered into the foot pad of three mice—two NZB mice and a BALB/c mouse—every two days for the first five doses and then weekly for three booster doses. Blood was collected on Days 0, 14, and 31 to determine the antibody concentration in the serum by ELISA. (**B**) Schematic depicting the general workflow for the generation of PANK3 antibody. Briefly, recombinant 20 µL human PANK3 protein and IFA mix was injected into the foot pad of three mice (two NZB and one BALB/c). Following eight injections, mice were euthanized and plasma cells from the spleen were isolated and were fused with murine myeloma cells, Sp20, to form fusion hybridoma cells. After selection in HAT medium, the antibody-producing plasma cells were single-cell cloned and media supernatant from each clone was subjected to ELISA screening. (**C**) Serum titration of anti-PANK3 antibody, collected before mice sacrifice on Day 31, was done using ELISA assay. Serum levels of PANK3 antibody was higher in mouse #2, which was used for subsequent hybridoma generation, single clone isolation, and antibody production. Serum response to His tag is compared to a blank.

**Figure 4 biomolecules-12-01323-f004:**
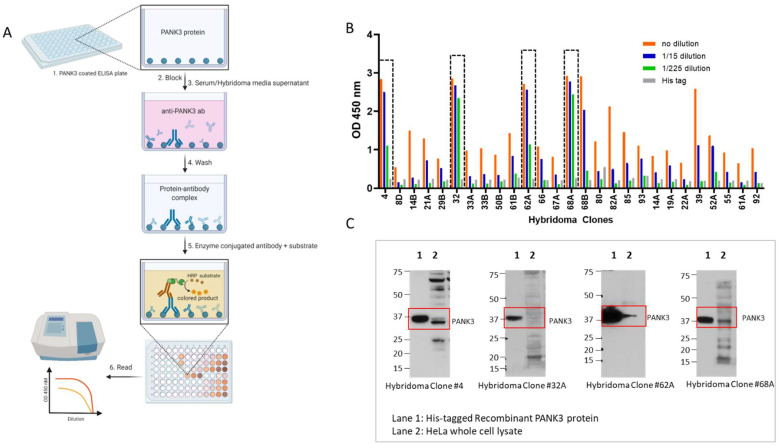
Quantification of PANK3 antibody levels in media supernatant of hybridoma clones. (**A**) Schematic showing the detection of PANK3 protein by ELISA. Briefly, ELISA plates were coated with recombinant PANK3 protein and media supernatant (primary antibody) isolated from different hybridoma clones or the serum extracted from the immunized mice were added to the wells. An HRP-linked secondary antibody was added to detect the PANK3 antibody bound to the immobilized PANK3 protein on the well. Addition of the substrate yielded a color change and the absorbance was quantified using a spectrophotometer. (**B**) Absorbance values of the medium supernatant (1:15 serial dilutions) isolated from different hybridoma clones. Only 14 clones out of 27 are shown in the graph (see Appendix A for all 27 clones). (**C**) Immunoblot showing the four antibody clones that were selected for further purification. The antibody clones were able to identify the recombinant protein, but the endogenous protein band in the HeLa cell lysate was convoluted by overlapping non-specific bands.

**Figure 5 biomolecules-12-01323-f005:**
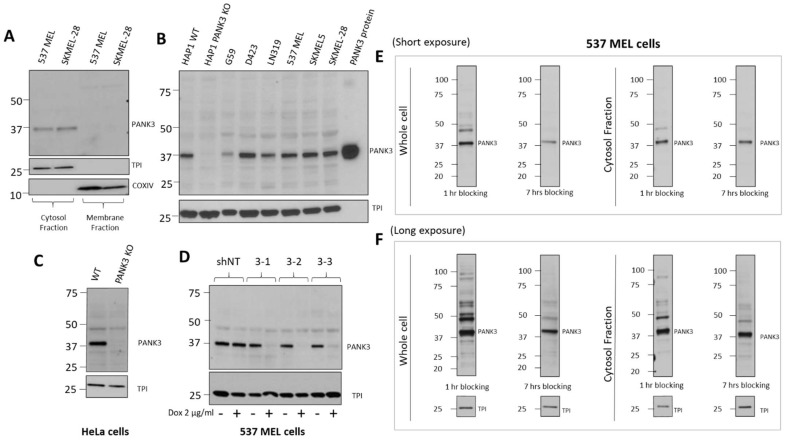
Optimization of PANK3 specific detection by subcellular fractionation of cells and longer blocking of transferred membranes. (**A**) Immunoblots showing that PANK3 protein is found exclusively in the cytosolic fraction, confirming the cytosolic distribution of PANK3 protein in human cells. B. MDA-299-62A can detect the recombinant PANK3 protein (positive control) and endogenous PANK3 protein in cytosolic fractions of specified cancer cell lines. HAP1 *PANK3* CRISPR KO (**B**) and HeLa *PANK3* KO (**C**) cell lines were used as negative controls to verify the correct band specificity. (**D**) Knockdown of PANK3 protein by dox-inducible shRNA, determined by immunoblot on cytosolic protein fractions using MDA-299-62A mouse monoclonal PANK3 antibody. (**E**,**F**) The PANK3 band was detectable on whole cell lysates and the interference by non-specific bands was significantly reduced by increasing the duration of blocking to 7 h instead of the typical 1 h. Both the whole cell lysates and cytosolic protein fractions are shown for reference, with shorter (15 s) and longer exposure (5 min) to the X-ray film.

## Data Availability

All data relevant to the manuscripts are included.

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
