# Peer review of "Generation and Validation of an Anti-Human PANK3 Mouse Monoclonal Antibody"

_biomolecules, 2022, doi:10.3390/biom12091323_

Round 1

Reviewer 1 Report

The manuscripts by Khadka S. et al describes the generation and characterization of an antibody for the detection of endogenous levels of human PANK3 in cancer cell lines. The manuscript is well written and the experiments carefully controlled. Additional comments are below.

Major

1)  The authors should include the expected (calculated based on the sequence) molecular weights for all the PANK isoforms, including PANK1a and PANK1b. The CST antibody seems to detect only Pank1b (41.5 kDa), as the band is below the 50 kDa marker line. However, PANK1a and 1b share the same sequence, with PANK1a having a longer N terminus, suggesting that none of the tested cell lines expressed PANK1a, which is odd but possible.  Have the authors verified whether any of these cell lines actually generates PANK1a transcripts?

2)    Does the antibody detect mouse PANK3? Confirming this feature using mouse tissue lysates would broaden the application and utility of the antibody generated by the authors.

Minor

E. coli” should be italicized throughout the text

Author Response

Please see attachment below

Reviewer 2 Report

The authors present the generation of a mouse monoclonal antibody against human PANK3 protein, which is not commercially available to date. The study is scientifically sounds. In particular, the western blotting combined with sub-cellular fractionation as well as the optimization of blotting condition are well done to validate the antibody to detect PANK3. The manuscript should be expanded to address the following concern. Preparation of homogeneous proteins is also important to obtain antibodies which recognize conformational epitope. It is not clear only from SDS-PAGE after the size exclusion chromatography (SEC) that the protein sample is homogeneous (monomer? dimer?) or not, and the SEC elution pattern should be shown to clarify this point.

Reviewer 3 Report

This manuscript deals with the generation of a specific PANK3 antibody. Lack of specific antibodies against the PANK proteins have indeed hampered progress in several fields of research and therefore generation of such an antibody is a most welcome addition to the toolbox of researchers working on PANK proteins.

After having carefully studied this manuscript, I have some remarks for the authors to improve their manuscript. As the work seems to have been executed with all the proper controls, all of the improvements are mostly textual in nature. Find below my comments in bullet point:

1.     The authors speak of PANK isoforms. Technically speaking PANK1, 2 and 3 are derived from duplications of an ancestral gene, currently encoded by separate genes and located on different parts of the genome. Therefore, they are paralogs, not isoforms. Only PANK1 alpha and beta are derived from the same gene and can be called isoforms. For biological correctness this needs to change throughout the manuscript.

2.     Although technically maybe not relevant for this manuscript, the authors do not mention PANK4. Because the authors find PANK proteins relevant as precision oncology collateral targets, 2 recent studies involving PANK4 (1.Dibble, C. C. et al. PI3K drives the de novo synthesis of coenzyme A from vitamin B5. Nature 1–7 (2022), and 2 .Li, X. et al. Pantothenate Kinase 4 Governs Lens Epithelial Fibrosis by Negatively Regulating PKM2 Related Glycolysis. Biorxiv 2022.08.02.502446 (2022).), may warrant mentioning these in the introduction or discussion.

3.     The authors mention in their introduction that inactivating mutations in PANK2 have been linked to PKAN in humans. To my surprise, THE article that makes that link for the first time and coins the term PKAN is not referenced: 1.Zhou, B. et al. A novel pantothenate kinase gene (PANK2) is defective in Hallervorden-Spatz syndrome. Nat Genet 28, 345–349 (2001). 

4.     In addition to the reference oversight above, it seems the reference list is heavily biased towards 1 specific research group. 16 out of 28 references share the same author. Although this person is important in the PANK field, the field is broader than this. I suggest updating the references to better reflect this.

5.     The authors mention PANK proteins as precision oncology collateral targets, since the PANK1 gene is co-deleted as part of the PTEN locus in some highly aggressive cancers. I suggest the authors consider rephrasing this suggestion. Now it is suggested in refs 21 and 22, that PANKs are proven collateral targets, because of PANK1 co-deletion. However, the proof of principle is given for another co-deleted gene, not PANK proteins. 

6.     The authors mention in their introduction: “Despite the obvious importance of PANKs in CoA biosynthesis, the lack of reliable antibodies that can detect and distinguish endogenous PANK isoform proteins, has significantly thwarted efforts to better understand their role in both normal physiology and in pathological contexts, as well as characterize them as therapeutic targets.” They go on to mention in the results and discussion that they found antibodies against PANK1 and 2 that are specific. While technically correct, the PANK2 antibody was already described to be specific in: 1.Lambrechts, R. A. et al.CoA‐dependent activation of mitochondrial acyl carrier protein links four neurodegenerative diseases. EMBO molecular medicine 109, 139–21 (2019), and 2.Dibble, C. C. et al. PI3K drives the de novo synthesis of coenzyme A from vitamin B5. Nature 1–7 (2022). In addition, the PANK1 antibody that the authors describe has also been described by Dibble et al. 

So, although an important verification of reproducible results, not a novelty perse.

7.     The text accompanying figure 5, states that the CRISPR exp. is consistent with large…as well as mouse germline KO studies. While that may be true, both refences (2 and 25) do not contain a PANK3 KO. Actually, one of them states that analysis of this line is progress. Please update with a proper reference or revise the sentence. 

Regarding the figures I have the following comments: 

8.     Figure 4B and S4 do not add significant different data. I suggest to replace figure 4B with figure S4. Then all the data is shown together. 

9.     Figure S6 refers to figure 5. Looking at the results, the right-hand side refers to figures 5D and E. I think this needs to be 5 E and F.

And my final comment: I noticed that the KO lines are commercially made available, but nothing was mentioned regarding the PANK3 antibody. Will this be commercially available as well, or the hybridomas themselves?
